# UWYN (Use Only What You Need): Efficient Inferencing in 2D and 3D Vision Transformers

## Abstract

Efficient computation for Vision Transformers (ViTs) is critical for latency-sensitive applications. However, early-exit schemes rely on auxiliary controllers that introduce non-trivial overhead. We propose UWYN, an end-to-end framework for image classification and shape classification tasks that embeds exit decisions directly within the transformer by reusing the classification head at each residual block. UWYN first partitions inputs via a lightweight feature-threshold into "simple" and "complex" samples: simple samples are routed to a shallow ResNet branch, while complex samples traverse the ViT and terminate as soon as their per-block confidence exceeds a preset confidence threshold. During the ViT pass, UWYN also dynamically prunes redundant patch embeddings and attention heads to further reduce computation. We implement and evaluate this strategy on both 2D (ImageNet, CIFAR-10, CIFAR-100, SVHN, BloodMNIST) and 3D (ModelNet-40, Scan Object NN) benchmarks. UWYN reduces Multiply-Accumulate operations (MACs) by over 75% compared to SOTA models such as LGViT (ACM MM '23), achieving 83.29% accuracy on CIFAR-100 and 84.39% on ImageNet. We also show faster inference with minimal accuracy loss.

## 1 Introduction

Vision Transformer (ViT) (Kolesnikov et al., 2021) architectures have emerged as powerful tools for a wide range of computer vision tasks. However, their high computational demands present challenges, especially in resource-constrained environments like mobile and embedded systems. These systems, equipped with powerful Systems-on-Chips and heterogeneous processing units, require models that balance accuracy with efficiency to meet real-time processing needs. Despite significant advances in accuracy, traditional ViT models often overlook the resource constraints, such as limited processing power, memory, and energy, faced by such devices. For instance, mobile GPUs typically have limited memory, ranging from 2-8 GB, which is far below the requirements of

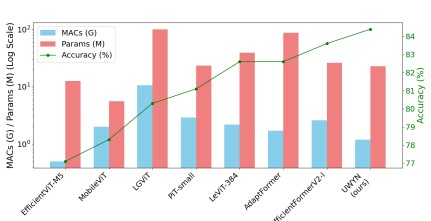

Figure 1: Computational complexity (MACs), model size (number of parameters), and ImageNet accuracy across different ViT methods (Liu et al., 2023; Mehta & Rastegari, 2022; Xu et al., 2023; Heo et al., 2021; Graham et al., 2021; Chen et al., 2022; Li et al., 2022b). UWYN substantially reduces MACs with only a minimal drop in accuracy.

large-scale transformer models. To illustrate, let us consider a state-of-the-art transformer model like MegaViT (Dehghani et al., 2023), which has 22 billion parameters. Since each parameter is typically stored as a 4 byte floating-point value, storing the model's parameters alone requires approximately 88 GB of memory. This large memory requirement renders models like MegaViT impractical for most GPUs or TPUs. Even high-end GPUs with 40-80 GB struggle to fit such models without optimizations like model parallelism, sharding, or offloading model segments to disk. Smaller network architectures like ResNet (He et al., 2016) rely on local convolutions, which limits their ability to capture long-range dependencies. In contrast, transformers address this by using self-attention mechanisms that model global relationships. While both ViTs and ResNets have demonstrated impressive capabilities, there is increasing interest in using early exits to improve efficiency and model interpretability. Recent work has focused on training smaller networks and applying algorithmic approximations to tailor transformer-based models for embedded devices (Chen et al., 2022; Xu et al., 2023). While

these approaches have made strides in reducing model size and computational demands, they often compromise accuracy or limit the model's full potential. This trade-off highlights the need for solutions that optimize ViTs for deployment in real-world, resource-constrained environments — solutions that can significantly reduce computational requirements without drastically reducing performance.

This forms our problem statement: *How can a ViT dynamically allocate computation based on input complexity?* Our solution is to approximate the execution depth based on content complexity, allowing the model to dynamically adapt its processing to each input. Consider a ViT architecture with 12 blocks, applied here to ImageNet classification (Deng et al., 2009; Steiner et al., 2021). To quantify the depth–efficiency trade-off, we systematically drop the final $k$ blocks from a 12-layer ViT and record end-to-end inference time versus classification accuracy, as shown in Table 1. Similar trends have been observed in NLP problems (Appendix A.8), highlighting a fundamental trade-off between accuracy and efficiency that can be exploited for early exits that allocate computation based on input content when rapid inference or resource constraints are critical.

Table 1: **Motivation:** Trade-off between inference time and accuracy on ImageNet when dropping the final $k$ of a 12-layer ViT.

| Number of Blocks Dropped | Inference Time (s) | Accuracy (%) |
|---|---|---|
| 0 | 287 | 84.8 |
| 1 | 264 | 82.5 |
| 2 | 242 | 65.2 |
| 3 | 217 | 34.6 |
| 4 | 197 | 12.0 |

The pipeline begins by classifying inputs by complexity: simple data goes to a shallow ResNet, complex data to a novel early-exit Transformer. Our model adds a layer within each Transformer block to leverage intermediate features, enabling context-aware, efficient processing by selecting key patches and attention heads. Embeddings from each block feed into a shared classifier (Figure 2). After each block, token embeddings are classified; if a dominant class exceeds a threshold, we exit early with that prediction. We evaluate using accuracy, MACs, and parameters (Figure 1). Existing adaptive ViTs like EfficientViT (Liu et al., 2023) implements Cascaded Group Attention to perform faster inference, while methods like EfficientFormer (Li et al., 2022b) uses meta blocks for efficient token mixing and reduce computation cost. Various state-of-the-art methods inflate models with extra networks requiring separate training. UWYN minimizes overhead by selecting key features within a block and using a shared, light layer for early exits across all blocks, reducing parameters and MACs. Our gains have been demonstrated in two distinct tasks, 2D image classification and 3D shape classification. We achieve orders of magnitude gains in MACs relative to ViT , which is an exact, non-approximated model with the highest accuracy. We reduce MACs from 1693G to 1.2G and the number of parameters by $3.8\times$, from 86M to 22.86M. Our results show that UWYN achieves competitive accuracy with significantly reduced inference time, establishing it as a practical and efficient inferencing solution for real-world applications on resource-constrained devices. The main contributions of our paper are:

1. **Assessing feature relevance within blocks for adaptive early exits:** We integrate lightweight layers inside transformer blocks to assess feature relevance in real time, enabling content-aware early exit *without* auxiliary networks, simplifying training.
2. **Reusing a unified classifier with confidence-based early exits:** We reuse a single classifier across all transformer blocks, avoiding separate per-block classifiers and simplifying the architecture. Further, we implement a confidence score-based mechanism that continuously evaluates prediction certainty and triggers an early exit once a calibrated confidence threshold is met.
3. **Performance gains across diverse hardware architectures:** We demonstrate inference speedups across both server-class (NVIDIA P100) and edge-class GPUs (Jetson AGX Xavier and Jetson Orin), indicating that UWYN translates to practical gains in realistic deployment settings.

## 2 RELATED WORK

**Transformer Efficiency Challenges and Architectural Solutions:** Transformers (Vaswani et al., 2017) have achieved strong performance in tasks like image classification but face challenges with latency and resource demands due to large model sizes and reliance on extensive datasets (Radford et al., 2021; Li et al., 2022a; 2023a; Koh et al., 2024). These limitations have driven research into adapting transformers for mobile and resource-limited environments (Li et al., 2022b; Wu et al., 2022; Zhang et al., 2022). Several architectural modifications have been proposed to address efficiency constraints. For instance, EfficientViT (Liu et al., 2023) introduces cascaded group attention blocks to process features selectively, avoiding full-length processing and reducing computational load.

Efficient-Former uses a dual path attention mechanism to reduce inference time (Li et al., 2022b). Similarly, MobileViT (Mehta & Rastegari, 2022) separates local and global feature processing with CNNs before integration, and AdaViT (Meng et al., 2022) uses a decision network in each block to focus on essential parts of prior outputs, optimizing efficiency. This adds an overhead to each transformer block, thereby making the overall process heavyweight. In addition to architectural changes, data flow optimization has been explored to improve transformer training efficiency. DeiT (Touvron et al., 2021a) integrates a distillation token and pre-trained CNN teacher to guide learning. PiT (Heo et al., 2021) uses depth-wise convolution to achieve channel multiplication and spatial reduction with small parameters. In video tasks, SparseVit (Chen et al., 2023b), a variant of Swin Transformer (Liu et al., 2021b), prioritizes high-relevance windows based on L2 norms, improving execution efficiency in self-attention mechanisms. However, by separating local and global attention, the number of parameters is increased significantly. Other orthogonal efforts have been made to efficiently handle memory processing for hardware, such as Ring Attention (Liu et al., 2024) and Flash Attention (Dao et al., 2022). However, these need powerful tensor cores, which makes it infeasible for mobile devices like AGX-class of mobile GPUs and even earlier server-class GPUs like Tesla P100. In addition, the software dependencies for these methods have a conflict based on their current CUDA versions and JetPack packages available. In contrast to all works in this category, UWYN demonstrates reduction in inference time over a wide variety of mobile devices.

**Pre-processing hinders Efficiency in 3D Shape Classification:** For point cloud data, methods like PointNet (Qi et al., 2017a) and PointNet++ (Qi et al., 2017b) introduce symmetry and hierarchical clustering for direct, permutation-invariant processing. While transformers improve generalization and long-range dependency capture in point cloud tasks (Zhao et al., 2021), these models often depend on extensive pre-processing, which can reduce inference speed and efficiency, particularly in complex or masked scenes (Uy et al., 2019). In contrast, UWYN identifies how simple and complex data instances can be processed separately, thereby making the overall pipeline more efficient. We add a pre-processing step for 3D data, which accelerates the pipeline much more compared to the existing methods. These pre-processing steps, combined with early exit, make the end-to-end inference faster.

**Early Exits and Lightweight Networks in Transformers:** To reduce processing time, early exit techniques in transformers and deep neural networks assess intermediate results, terminating processing once a confidence threshold is met. Methods like those by Teerapittayanon & McDanel (2016) and Wołczyk et al. (2021) incorporated a small network for each network block or pathway to decide when to early exit. LGViT (Xu et al., 2023) achieves this by combining local and global perception heads, while models like FastBERT (Liu et al., 2020) and BERxiT (Xin et al., 2021) add lightweight networks within BERT layers for early exits. However, these approaches generally increase computational complexity by requiring additional networks or classifiers. Techniques like Neural Architecture Search (NAS) (Li et al., 2023c; Zoph & Le, 2017; Yang et al., 2023b) implement reinforcement learning to decide when to exit early or which pathway to choose to streamline computation. These methods incur significant computational overhead due to the addition

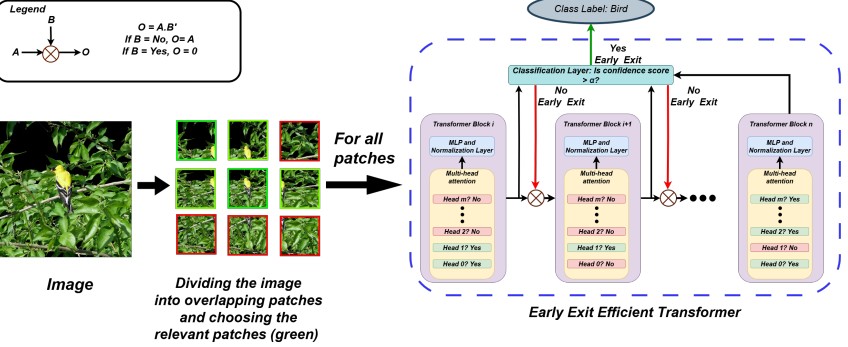

Figure 2: **Architecture of Early-Exit Efficient Vision Transformer:** The input image is segmented into overlapping patches by the Soft Split Network, analyzed by the Patch and Attention Head Selector to identify key embeddings and attention heads. Each modified transformer block includes normalization, multi-head attention, MLPs, and a mechanism for selecting relevant attention heads. Embeddings are evaluated at a classifier layer; if the confidence score exceeds $\alpha$ or the final block is reached, classification is returned. Otherwise, processing continues to the next transformer block, reducing redundant computation.

of separate auxiliary networks at multiple layers of the backbone. We quantitatively compare the

efficiency and the accuracy of UWYN against several of these prior works: ViT, EfficientViT, LeViT, EfficientFormer, PiT, MobileViT, AdaptFormer, and LGViT in Table 2.

## 3 METHODS

As shown in Table 1, reducing the number of blocks in the ViT architecture results in substantial computational savings with only a slight decrease in accuracy. For instance, using 90% of the model maintains classification accuracy while achieving a speedup of approximately 20 seconds in inference time (10% less time). This observation provides the insight that the full model is not necessary for confidently classifying all data instances. To leverage this insight, we propose a systematic approach for accelerating inference by categorizing data based on its complexity and adapting processing accordingly.

Our proposed pipeline, illustrated in Figure 3, divides input data into two categories: A) simple data, which can be accurately classified with a simple network like ResNet and B) complex data, which requires heavy computations to be performed by an efficient early-exit ViT. We first demonstrate this approach on 2D image classification and then extend it to a 3D task with only minimal modifications to the pre-processing stage.

### 3.1 THRESHOLDING

Figure 3: **Flowchart for image classification using UWYN:** A dual-path strategy based on complexity: simple images are routed to a lightweight residual network, while complex ones pass through an early-exit transformer where a shared classifier checks the confidence score $\alpha$ at each block, stopping when the threshold is met.

Data samples with complexity below a threshold are routed to the simple branch; rest follow the complex path.

**2D images:** We measure image complexity with the Sobel operator (Appendix A.1), which computes gradient magnitude by convolving the image with two 3×3 kernels for horizontal $G_x$ and vertical edges $G_y$. The gradient magnitude is:

$$\text{Magnitude}(i, j) = \sqrt{G_x(i,j)^2 + G_y(i,j)^2}. \tag{1}$$

We aggregate the gradient magnitude over all pixels to obtain a scalar complexity score per image. Using an empirically set threshold (Section 4.1), we classify images in the top $2/3^{\text{rd}}$ of gradient magnitudes as complex and the rest simple. Simple images have fewer edges and are easier to classify, indicating less structural detail, while complex images contain richer edge information. In Figure 7 (Appendix A.2), we illustrate this split on ImageNet. Because the Sobel operator relies on fast gradient computations, our complexity score runs efficiently and fits resource-constrained settings.

**For 3D:** We represent each shape as a point cloud of 1,024 points in $(x, y, z)$ coordinates and estimate complexity from the volume these points occupy. We classify shapes in the lower $70^{\text{th}}$ percentile of volume (denser, more detailed objects) as complex and treat the rest as simple. We choose this volume threshold empirically.

### 3.2 SIMPLE NETWORK

**For 2D.** We use ResNet50 (He et al., 2016) to process the simple data. We train the network from scratch with drop-path regularization (Huang et al., 2016), which improves robustness and generalization. This simple branch speeds up the overall pipeline by handling low-complexity images with far fewer compute resources.

**For 3D.** We treat each point $(x, y, z)$ in the point cloud as a 3-channel input and use a ResNet-style architecture with 3, 4, 4, and 3 residual blocks. This model efficiently processes simple 3D shapes without invoking the transformer.

### 3.3 EARLY-EXIT EFFICIENT TRANSFORMER

For complex data instances, UWYN routes the data to an early-exit Vision Transformer inspired by efficient attention (Figure 2). After the Soft Split Network extracts overlapping patches, the transformer processes only the selected key patches. At each block, we apply a unified classifier to all selected patch embeddings. If the confidence score passes a threshold, or the final block is reached, we exit early and return the prediction.

#### 3.3.1 ADDITIONAL PRE-PROCESSING FOR 3D

We render each point cloud by centering and scaling its $(x, y, z)$ coordinates and projecting them with a virtual camera. Depth $(z)$ controls the perceived size, so distant points appear smaller. For a point $P(x, y, z)$, the projected coordinates are:

$$x' = \frac{x}{z} \cdot d, \quad y' = \frac{y}{z} \cdot d \tag{2}$$

where $d$ is the distance from the camera to the projection plane. This scaling preserves the perspective, with distant points shrinking as their $z$-coordinate increases, effectively transforming the 3D coordinates into 2D with realistic depth perception.

#### 3.3.2 EXTRACTION OF PATCHES

We use a Soft Split Network (SSN) (Yuan et al., 2021) to segment $\mathbf{X} \in \mathbb{R}^{h \times w \times c}$ into *overlapping patches*, emulating a convolution operation with receptive fields. Patches of size $t \times t$ overlap by $o$ pixels, with optional padding $p$. This preserves continuity and captures local detail better than disjoint tokens. After flattening the patches, the data is transformed into a format compatible with subsequent transformer blocks. The number of generated tokens, $N$, is computed as $N = \left\lfloor \frac{h+2p-t}{t-o} + 1 \right\rfloor \times \left\lfloor \frac{w+2p-t}{t-o} + 1 \right\rfloor$, where each token is a flattened vector of size $d = ct^2$.

The SSN output, $\mathbf{X}_{\text{SSN}} \in \mathbb{R}^{N \times d}$, is then prepended with a classification token [CLS], $\mathbf{X}_{\text{cls}}$, to form the final embedding input, $\mathbf{X}_{\text{emb}} = [\mathbf{X}_{\text{CLS}}; \mathbf{X}_{\text{SSN}}], \quad \mathbf{X}_{\text{emb}} \in \mathbb{R}^{(N+1) \times d}$.

#### 3.3.3 PATCH SELECTOR AND ATTENTION HEAD SELECTOR NETWORK

To improve computation efficiency (Figure 2), we introduce a Patch Selector before the first transformer block, and an Attention Head Selector in each of the transformer blocks. These modules retain only the most relevant patches and attention heads; we zero out the rest, so they no longer contribute to the computation.

Each selector uses a linear layer followed by a sigmoid activation. Intuitively, we select a patch if it likely contains part of the target object. Formally, the Patch Selector is:

$$\mathbf{S}_{\text{patch}} = \sigma(W_{\text{patch}} \cdot \text{vec}(\mathbf{X}_{\text{SSN}})), \quad W_{\text{patch}} \in \mathbb{R}^{N \times Nd} \tag{3}$$

where $\cdot$ is matrix multiplication, $\text{vec}(\mathbf{X}_{\text{SSN}}) \in \mathbb{R}^{Nd}$ is the flattened version of $\mathbf{X}_{\text{SSN}}$, and $\sigma(\cdot)$ is the *sigmoid* function which is applied element-wise. We process all patches concurrently and select those with scores $\geq 0.5$. For non-essential patches, we set the corresponding tokens to zero, preserving tensor dimensions while skipping their computation.

A tensor $X_{\text{emb}} = [\mathbf{X}_{\text{CLS}}; \mathbf{X}_{\text{SSN}}] \in \mathbb{R}^{(N+1) \times d}$ is passed to the first transformer block as input. Thus, each patch generates one token plus there is the [CLS] token, which together constitute the input to the transformer component. The Attention Head Selector has a similar architecture:

$$\mathbf{S}_{\text{AttHead}}^{(l)} = \sigma(W_{\text{AttHead}}^{(l)} \cdot \text{vec}(\mathbf{X}_{\text{emb}}^{(l-1)})), \quad W_{\text{patch}}^{(l)} \in \mathbb{R}^{H \times (N+1)d'}, \mathbf{X}_{\text{emb}}^{(l-1)} \in \mathbb{R}^{(N+1) \times d'}. \tag{4}$$

With a threshold of $0.7$ (Figure 6), we select high-impact attention heads and use them to form the features $\mathbf{X}_{\text{emb}}^{(l-1)}$ passed to the next block $l$; these heads intuitively focus on image regions most relevant to the final classification.

**Parallelization of Attention Calculation - Efficient calculation of Query and Keys:** Standard self-attention e.g., PyTorch Developers (2022), computes query $\mathbf{Q}$ and key $\mathbf{K}$ separately via matrix

multiplications with input $\mathbf{X}$, requiring two fetches of $\mathbf{X}$ and sequential computation, increasing memory access overhead and inference time due to GPU memory bandwidth limits and serialization. UWYN computes $\mathbf{Q}$ and $\mathbf{K}$ simultaneously in one inference pass. We concatenate trained weight matrices $\mathbf{W_Q}$ and $\mathbf{W_K}$ into $\mathbf{W} = [\mathbf{W_Q}, \mathbf{W_K}] \in \mathbb{R}^{d \times 2d}$. Then, $\mathbf{XW} = [\mathbf{Q}, \mathbf{K}]$ is computed. This reduces inference time by fetching $\mathbf{X}$ once and parallelizing $\mathbf{Q}$ and $\mathbf{K}$ computation. This does not cause any change to the training process as the attention values are not affected by this optimization. The speedup for this parallelization has been demonstrated in Appendix A.3.

### 3.3.4 Confidence Score and Early-Exit

We introduce two techniques to speed up processing within the transformer: *patch-wise prediction* and *early exit mechanism*.

**Patch-wise prediction:** For image and object classification datasets, where each image prominently contains a single object, the division of images into patches generally results in most patches reflecting the same object class. To exploit this characteristic, we extend beyond the conventional reliance on the single [CLS] token for classification, and propose prediction based on *all* the patches. Specifically, with the output of a transformer block $l$, i.e., the embedding features $\mathbf{X}_{\text{emb}}^{(l)}$, we have a universal classifier applied onto each patch embedding:

$$p_{\text{i}} = \arg\max(\text{softmax}(W \cdot \mathbf{X}_{\text{emb},i}^{(l)})), \quad W \in \mathbb{R}^{C \times d}, \forall i = 0, 1, \cdots, N. \tag{5}$$

In this paper, we implement a one-layer classifier. Among the $N + 1$ patch predictions $p_{\text{i}}$, the most frequently predicted class is marked as the image prediction $p = mode(p_{\text{i}})$. We compute a *confidence score* $\alpha = population(p)/(N + 1)$, which is the ratio of the votes for $p$.

**Early exit mechanism:** After each transformer block, we process the embedding features $\mathbf{X}_{\text{emb}}^{(l)}$ using the universal classifier (Eq. 5), and the patch-wise prediction strategy for classification. When a confidence score $\alpha$ surpasses a predefined threshold $\alpha_{\text{threshold}}$, indicating a strong consensus among the patch predictions, an early exit of the transformer model is executed: the prediction based on the intermediate embedding features is returned as the final output, and no further computation by subsequent transformer blocks is executed, hence saving computation. Note that this classifier, including weights, is shared by all tokens and all transformer blocks. By reusing the same classifier, we reduce the training overhead; by keeping our classifier lightweight, we reduce the inference time. This training time and inference time advantage over prior works like EfficientViT and EfficientFormer is demonstrated in Tables 2 and 3.

## 4 Results

We evaluate UWYN on 2D and 3D classification tasks, comparing its performance with state-of-the-art models in terms of accuracy, inference time, and computational efficiency.

**Datasets and training:** We evaluate our pipeline on ImageNet (Deng et al., 2009), CIFAR-10 (Krizhevsky & Hinton, 2009a), and CIFAR-100 (Krizhevsky & Hinton, 2009b). We also show our results on two unconventional datasets, BloodMNIST (Acevedo et al., 2020; Yang et al., 2021)—a low resolution medical image processing dataset, and SVHN (Netzer et al., 2011)—an image dataset, which includes potential distractors. For 3D shape classification, we have used ModelNet-40 (Wu et al., 2015) and Scan Object NN (Uy et al., 2019), where the latter includes distractions beyond the intended shape. We employ the Adam optimizer (Kingma & Ba, 2015) with an initial learning rate (with cosine annealing) of 0.1, $\beta = (0.9, 0.99)$, decayed by 1e-4 for the first 100 epochs, followed by a reduced rate of (2e-5) until convergence.

For our Soft Split Network (Section 3.3.2), we use three convolutional layers with kernel sizes $(7, 3, 3)$, strides $(4, 2, 2)$, and padding $(2, 1, 1)$, respectively. This configuration divides a $224 \times 224$ image into $14 \times 14$ patches. The Early-Exit Efficient Transformer has 12 blocks total, each block having 6 attention heads. The hidden dimension for ImageNet, ScanObjNN, and ModelNet40 is 768, while for other datasets, it is 384.

**Evaluation metrics:** We run each configuration five times on each device, with no other workloads running, and report averaged results. We measure top-1 image or shape classification accuracy,

Table 2: **ImageNet classification on edge and server GPUs.** Top-1 accuracy, MACs, parameters, and end-to-end inference time on Jetson AGX Xavier, Tesla P100, and Jetson Orin. UWYN achieves 84.39% accuracy with only 1.2G MACs and 22.9M parameters, over 90× fewer MACs than ViT-Base, and runs up to 2.3× faster than EfficientFormerV2-l (see Appendix Table 7 for additional model variants).

| Method | MACs | Params | Accuracy | Time [Xavier] | Time [Orin] | Time [P100] |
|---|---|---|---|---|---|---|
| ViT-Base (Tseng et al., 2022) (non-approximated model) | 1693 G | 86 M | 84.8% | 4960 s | 4482 s | 1089.52 s |
| EfficientViT-M5 (Liu et al., 2023) | **0.5 G** | 12.57 M | 77.1% | 3142.23 s | 3246.74 s | 792.63 s |
| LeViT-384 (Graham et al., 2021) | 2.2 G | 39.11 M | 82.6% | 1842.81 s | 1934.33 s | 511.8 s |
| EfficientFormerV2-l (Li et al., 2023b) | 2.6 G | 26.3 M | 83.6% | 4162 s | 4347.74 s | 1091.99 s |
| PiT-small (Heo et al., 2021) | 2.9 G | 23.5 M | 81.1% | 1976.98 s | 1919.21 s | 502.6 s |
| MobileViT (Mehta & Rastegari, 2022) | 2.0 G | **5.6 M** | 78.3% | 1809.65 s | 1747.32 s | 497.74 s |
| MobileViT-v2 (Mehta & Rastegari, 2023) | 2.0 G | 10.6 M | 80.4% | 1801.65 s | 1733.49 s | 492.86 s |
| AdaptFormer (Chen et al., 2022) | 1.71 G | 87.6 M | 82.6% | 3256.45 s | 3129.56 s | 719.82 s |
| LGViT (Xu et al., 2023) | 10.65 G | 101 M | 80.3% | 2001.9 s | 1984.5 s | 504.89 s |
| PartialFormer (Vo et al., 2024) | 3.4 G | 64 M | 83.9% | 1955 s | 1864 s | 498.42 s |
| FastViT-MA36 (Vasu et al., 2023) | 8.85 G | 42.7 M | 84.9% | 3641.22 s | 3422.93 s | 802.9 s |
| FasterViT-2 (Hatamizadeh et al., 2024) | 4.35 G | 75.9 M | 84.2% | 3629.39 s | 3441.07 s | 823.44 s |
| RepViT-M2.3 (Wang et al., 2024) | 4.5 G | 22.9 M | 83.7% | 2019.62 s | 1863.22 s | 599.82 |
| UWYN | 1.2 G | 22.86 M | **84.39%** | **1796 s** | **1695.23 s** | **492.63 s** |

average MACs (Multiply–Accumulate operations), number of parameters (Params), and total inference time over the full test set. We use the Python `thop` library to compute MACs and Params. The variation in inference time across the five runs is at most 7%. We report inference time on a server-class GPU (Tesla P100) and two resource-constrained edge devices: NVIDIA Jetson AGX Xavier and NVIDIA Jetson Orin.

## 4.1  2D RESULTS

We focus on single object classification tasks. Tables 2, 3, and 7 (Appendix) show how UWYN performs with respect to other state-of-the-art algorithms. We choose ViT as the baseline for its non-approximated structure and highest reported accuracy. We report the baselines' results based on their official implementations from GitHub and where available, pre-trained models from Hugging Face; we executed them ourselves on our target device types.

**ImageNet Results:** Table 2 demonstrates our performance on ImageNet over various state-of-the-art popular approximation methods. We compare UWYN with the highest accuracy for a given model, where it has several variants. For a comparison with other shallower versions of these models, please refer to Table 7 in Appendix. While there are models that have lower MACs (*e.g.,* EfficientViT) or fewer parameters (*e.g.,* MobileViT) individually, UWYN is consistently a better tradeoff between the MACs and number of parameters. Apart from the vanilla version of ViT (which is an exact, *i.e.,*, non-approximated model), UWYN demonstrates higher accuracy than other approximation methods. We further perform inference at almost half the amount of time required by EfficientViT and EfficientFormer. The fact that UWYN has lower values for MACs and Params emphasizes the efficiency of our model, achieving reduced computational costs without a significant sacrifice in performance. As compared to AdaptFormer, UWYN uses 40% lower MACs, 25% of the number of parameters, yet outperforms it in accuracy with lesser inference time on all devices.

**Unconventional 2D datasets and other popular datasets:** For 2D image classification, we evaluate on unconventional datasets, such as low resolution images like BloodMNIST and noisy images like SVHN. From the results in Table 3, we see that UWYN achieves competitive accuracy, maintaining a low inference time compared to traditional models like ViT and LGViT. It is worth noting that the BloodMNIST website (Yang et al., 2023a) reports achieving a 96.1% accuracy using Google AutoML Vision, with 2 hours of node time required for training. In contrast, training the ResNet model and the Early-Exit Transformer for our approach took only 20 and 100 epochs, respectively, amounting to approximately 1 hour of total training time from scratch. Our method UWYN achieved a 96.17% accuracy. However, since they have not released their code, it is difficult to directly compare the speedup. AdaptFormer achieves the highest accuracy on CIFAR100 (86.2%) with an inference time of 136.67 s. UWYN demonstrates a balance of efficiency and accuracy across CIFAR100 (83.29%, 103.63 s), CIFAR10 (94.87%, 103.4 s), BloodMNIST (96.17%, 27.88 s), and SVHN (92.16%, 225.67 s), often with fewer MACs and parameters. UWYN performs inference faster consistently over all three device classes — server class GPU as P100, and edge devices as AGX Xavier and Jetson Orin.

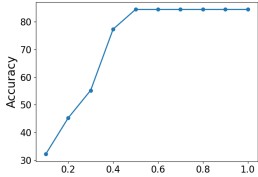 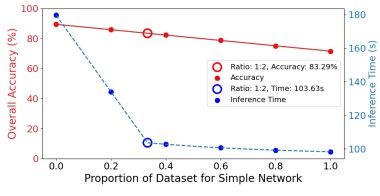 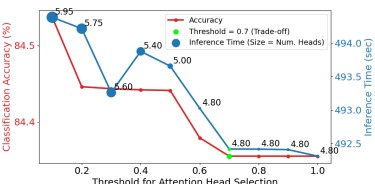

Figure 4: ImageNet accuracy vs. majority-vote threshold $\alpha_{\text{threshold}}$: from 32.5% at [$\alpha$=0.1] to 84.39% at [$\alpha$=0.5], then plateauing.

Figure 5: Accuracy [red, left] and inference time [blue, right] vs. fraction routed to the simple branch on CIFAR-100; at 33% simple, UWYN achieves 83.29% accuracy in 103.63 s, a reasonable point in the speed-accuracy trade-off.

Figure 6: Varying the threshold for attention head selection for ImageNet classification task. A threshold value of 0.7 strikes a reasonable trade-off between the accuracy and the inference time.

Table 3: **Variety of Datasets:** Performance on 2D image classification tasks across various datasets, showing accuracy and inference time on P100.

| Method [2D] | CIFAR100 | | CIFAR 10 | | BloodMNIST | | SVHN | |
|---|---|---|---|---|---|---|---|---|
| | Accuracy | Time | Accuracy | Time | Accuracy | Time | Accuracy | Time |
| ViT | 93.95% | 219.47 s | 95.29% (cif) | 219.52 s | 89.6% (vit) | 58.68 s | 96.40% (svh) | 275.8 s |
| LGViT | 82.57% | 94.98 s | 92.5% | 95.27 s | 91.5% | 30.28 s | 90.24% | 250.66 s |
| AdaptFormer | 86.2% | 136.67 | 93.36% | 137.12 | 90.6% | 42.5 s | 93.2% | 230.56 s |
| UWYN | 83.29% | 103.63 s | 94.87% | 103.4 s | 96.17% | 27.88 s | 92.16% | 225.67 s |

**Choice of Hyper-parameters:** We observed that maximum accuracy was achieved at $\alpha_{\text{threshold}} = 0.5$ (equivalent to the majority case) when increasing the threshold $\alpha_{\text{threshold}}$ from 0.1 to 0.5 (Figure 4). Beyond this point, increasing $\alpha_{\text{threshold}}$ resulted in only marginal gains in accuracy (in the second decimal place). Therefore, an early exit is triggered if the confidence score exceeds the predefined threshold of $\alpha_{\text{threshold}} = 0.5$ (or if the majority of token predictions agree on the same object), and has been maintained throughout the results for the purpose of this paper.

We also explored the impact of varying the proportions of simple and complex data instances. We notice a similar trend of all datasets, and the results for CIFAR 100 are presented in Figure 5. Processing all data through the simple network yields the lowest inference time but also the lowest accuracy. Conversely, utilizing only the complex network results in high accuracy at the cost of significantly increased inference time. However, a closer examination of the graph reveals a notable trend: If the ratio of simple data instances is less than 0.3, the accuracy is high but the inference time is unacceptably high. There is a knee in the curve at 0.3 and beyond that, the inference time drops slowly, as does the accuracy. So, we choose 33.33% of the data points for all our experiments, as going the simple ResNet path. This reduction can likely be attributed to the increased prevalence of early exits in the complex network beyond this ratio, leading to faster processing, especially when combined with the efficient processing of the remaining simple instances by the simple network. This observed behavior strongly motivated our decision to adopt a 1:2 ratio for simple to complex data instances throughout our evaluation. For selecting the number of attention heads, the threshold of 0.7 empirically strikes a favorable balance. It maintains a high classification accuracy while achieving a noticeable reduction in inference time compared to lower threshold values, as observed in Figure 6. The numbers overlaid on the inference time data points refer to the number of attention heads used on average at a particular threshold. There are 6 total attention heads, and we observe that after a threshold of 0.6, an average of 4.8 attention heads are used.

## 4.2 3D RESULTS

Table 4 presents the performance of UWYN on 3D shape classification task using the ModelNet40 and ScanObjNN datasets. Our method outperforms existing approaches in both accuracy and inference time, such as Point Transformer (Zhao et al., 2021) and PointNet++ (Qi et al., 2017b). Specifically, UWYN achieves higher accuracy on both datasets—94.03% on ModelNet40 and 92.43% on ScanObjNN—while significantly reducing inference times (78.31% reduction on ModelNet40 and 73.4% reduction on ScanObjNN inference time with respect to PointTransformer (Zhao et al., 2021) on P100). These results highlight the efficiency of UWYN's approach, which reduces the dimensionality of point clouds and accelerates the processing pipeline compared to traditional methods. We repeat experiments on our edge devices and observe a speed up in inference time for all

cases. As discussed earlier, in Section 2, existing pipelines are slow primarily due to their need for heavyweight pre-processing; however, UWYN overcomes this limitation through a simple conversion to 2D data format. It is worth noting that our speed up in the 3D task relative to state-of-the-art is better than in the 2D task; this is attributable to the reason just given above.

Table 4: **3D shape classification on ModelNet40 and ScanObjNN :** UWYN achieves 94.03% and 92.43% accuracy while reducing inference time by over 75% on P100, AGX Xavier, and Orin compared to Point Transformer and PointNet++.

| Method [3D] | Accuracy | ModelNet40 Inference Time | | | Accuracy | ScanObjNN Inference Time | | |
|---|---|---|---|---|---|---|---|---|
| | | Xavier | Orin | P100 | | Xavier | Orin | P100 |
| Point Transformer | 92.4% | 3892.23 s | 3695.62 s | 842.41 s | 90.5% | 900.25 s | 821.45 s | 171.2 |
| PointNet++ | 91.8% | 2896 s | 2667.34 s | 705.04 s | 84.2% | 825.23 s | 798.66 s | 142.5 |
| UWYN | **94.03%** | **503.5 s** | **440.12 s** | **182.7 s** | **92.43%** | **150.27 s** | **139.55 s** | **45.45 s** |

### 4.3 OBJECT DETECTION RESULTS

In this section, we evaluate UWYN on object detection using an SSD (Liu et al., 2016) backbone and report results on the COCO dataset (Lin et al., 2014). We measure performance in terms of mean average precision [mAP] across all classes and compare against models of comparable scale as well as the strongest variants within each family. As shown in Table 5, UWYN delivers the highest accuracy

Table 5: **Object Detection Performance:** UWYN performs with a higher mAP using less than half of the number of parameters.

| Method | MACs [G] | Params [M] | mAP [%] |
|---|---|---|---|
| RT-DETRv2S (Lv et al., 2024) | 30 | 20 | 48.1 |
| RT-DETR-R18 (Zhao et al., 2024) | 30 | 20 | 46.5 |
| EfficientDet (Tan et al., 2020) | 99.6 | 64 | 59.9 |
| YOLO-12m (Tian et al., 2025) | 33.75 | 20.2 | 52.5 |
| YOLO-v8s (Jocher et al., 2023) | 14.4 | 11.16 | 43.2 |
| SSD (Liu et al., 2016)+UWYN | **10.84** | **4.31** | **64.22** |

while requiring less than half the number of parameters. Notably, UWYN achieves a 20% improvement in mAP over RT-DETRv2 (Lv et al., 2024) while using under 30% of its trainable parameters, demonstrating both the efficiency and extensibility of our approach across diverse problem settings.

## 5 CONCLUSION AND BROADER IMPACT

In conclusion, we introduced UWYN, a dynamic early-exit framework that enhances the efficiency of classification by explicitly targeting redundant computation. UWYN combines a content-aware routing scheme (simple vs. complex samples), a Soft Split patch extractor with patch and head selectors, and a shared confidence-based classifier to adapt depth and width on the fly. Quantitatively, reduces MACs by up to 25% relative to comparable efficient ViT methodologies while maintaining competitive or better accuracy, and delivers consistent speedups on both edge and server GPUs across 2D and 3D benchmarks. These results highlight the potential of UWYN as a viable and efficient solution for vision model deployment in resource-constrained real-world applications, including edge devices and unmanned aerial vehicles, where strict latency and energy budgets often preclude conventional ViTs. Beyond raw efficiency, UWYN offers a generic template for content-adaptive computation in transformers: it requires no task-specific auxiliary networks and can integrate with existing ViT backbones as a drop-in modification. Future work includes extending UWYN to other modalities (e.g., video, multimodal inputs) and co-designing it with emerging hardware for even tighter compute–energy trade-offs.

**Reproducibility statement:** We present training protocols, architectural specifications, and hyper-parameter choices in the Results section (Section 4). The underlying mathematical formulations are described in the Methods section (Section 3), while implementation details related to Sobel operators are provided in the Appendix (Section A.1). The source code will be released publicly upon acceptance of this paper.

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
