# A APPENDIX

Here, we will demonstrate the additional results from our work.

## A.1 SOBEL OPERATOR

The Sobel operator calculates the gradient of an image by convolving the image with two 3x3 kernels: one for detecting horizontal edges ($G_x$) and one for detecting vertical edges ($G_y$). The convolution of the image $I$ at each pixel $(i, j)$ with these kernels is expressed as:

$$G_x(i, j) = \sum_{m=-1}^{1} \sum_{n=-1}^{1} G_x(m, n) \cdot I(i + m, j + n) \tag{6}$$

$$G_y(i, j) = \sum_{m=-1}^{1} \sum_{n=-1}^{1} G_y(m, n) \cdot I(i + m, j + n) \tag{7}$$

where the Sobel kernels are defined as:

$$G_x = \begin{bmatrix} -1 & 0 & 1 \\ -2 & 0 & 2 \\ -1 & 0 & 1 \end{bmatrix}, \quad G_y = \begin{bmatrix} -1 & -2 & -1 \\ 0 & 0 & 0 \\ 1 & 2 & 1 \end{bmatrix} \tag{8}$$

The gradient magnitude is then calculated as:

$$\text{Magnitude}(i, j) = \sqrt{G_x(i, j)^2 + G_y(i, j)^2} \tag{9}$$

The gradient magnitude is a measure of the sharpness of the image - a measure of how much curvature or information is there in the image.

## A.2 VISUAL REPRESENTATION OF SIMPLE AND COMPLEX IMAGES

Figure 7 is a visual representation of a simple and a complex image from the ImageNet dataset. We identify few curves on the simple image and more detailed silhouettes on complex images, based on our thesholding mechanism in Section 3.1.

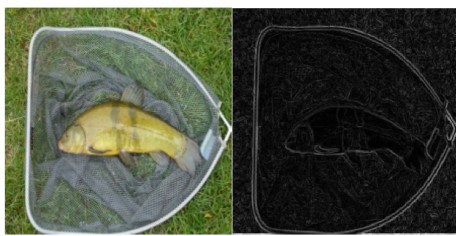 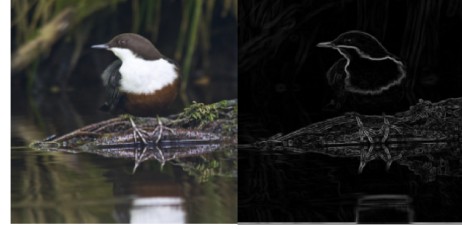

**(a) Simple Image Thresholding**  **(b) Complex Image Thresholding**

Figure 7: Examples from the ImageNet dataset Deng et al. (2009) illustrating the application of a Sobel operator [ A.1] for edge detection. (a) A "simple" image (left) with minimal details, processed with the Sobel operator (right), showing fewer prominent edges. (b) A "complex" image (right) with intricate details, processed with a Sobel operator (right), revealing a dense edge pattern.

## A.3 SPEED-UP DUE TO ATTENTION PARALLELIZATION

As discussed in Section 3.3.3, parallelizing attention computations significantly accelerates inference in the Efficient Early-Exit Transformer, with a more pronounced effect on 2D tasks than 3D. In 3D tasks, the extra pre-processing for point cloud data and 2D projection introduces additional computational overhead, limiting the speedup.

Table 6: **Parallel attention computation speed-up:** Comparison of data instances processed per second for Early-Exit Efficient Transformer, evaluated under both parallel and sequential attention schemes. The speedup in 2D images is more than 3D shapes.

| Dataset | Data instances processed per second | |
|---|---|---|
| | Attention parallel | Attention sequential |
| CIFAR 100 (Krizhevsky & Hinton, 2009b) | 60.15 | 58.37 |
| ImageNet (Deng et al., 2009) | 28.23 | 26.40 |
| ModelNet40 (Wu et al., 2015) | 13.74 | 13.57 |

## A.4 COMPARISON WITH EXTREMELY SMALL MODELS

Despite the existence of models with considerably lower MACs in Table 7, our evaluation reveals a clear advantage for our approach. Our model exhibits faster inference times, notably on edge devices, and maintains a higher accuracy than any of the identified low-MAC models. These experiments have been performed on ImageNet. We also add entries from recent models whose implementation is not publicly available at the time of submission. We use the symbol: − to indicate observations that are not accessible due to the aforementioned reason.

Table 7: **Comparison with other models:** Comparison of small MAC Methods with UWYN on ImageNet. We demonstrate higher accuracy than all and faster inference for some.

| Method | MACs | Params | Accuracy | Time (Xavier) | Time (P100) | Time (Orin) |
|---|---|---|---|---|---|---|
| Efficient ViT-M0 (Liu et al., 2023) | 0.1 G | 5.4 M | 71.9% | 1415 s | 346.87 s | 1279.38 s |
| LeViT-128S (Graham et al., 2021; Wightman, 2019) | 0.2 G | 7.77 M | 76.5% | 1582.14 s | 401.16 s | 1632.71 s |
| EfficientFormerV2-s0 (Li et al., 2023b; Wightman, 2019) | 0.3 G | 3.5 M | 76.1% | 1994.42 s | 494.13 s | 2068.05 s |
| LF-ViT (Hu et al., 2024) | 1.85 G | - | 82.2% | - | - | - |
| CF-ViT (Chen et al., 2023a) | 2.4 G | - | 81.9% | - | - | - |
| SAC-ViT (Hu et al., 2025) | 1.6 G | - | 82.3% | - | - | - |
| FastViT-SA24 (Vasu et al., 2023) | 1.9 G | 20.6 M | 82.6% | 2289.45 s | 594.66 s | 2397.85 s |
| FasterViT-0 (Hatamizadeh et al., 2024) | 1.65 G | 31.4 M | 82.1% | 2543.12 s | 659.23 s | 2895.2 s |
| RepViT-M1.1 (Wang et al., 2024) | 1.3 G | 8.2 M | 81.2% | 1524.22 s | 425.29 s | 1782.05 s |
| UWYN | 1.2 G | 22.86 M | 84.39% | 1796 s | 492.63 s | 1695.23 s |

## A.5 USING A LARGER BATCH SIZE TO COMPARE RESULTS

Here we compare the inference time when we use a batch size of 32 during inference time. There is a significant speedup for the 2D processing (Table 9). There is a larger speedup in the 3D pre-processing due to efficient handling of data as compared to the other methods (Table 8). We identify the simple and complex data instances from beforehand, batch them together and then perform inference.

Table 8: **Batch inference:** This table shows the time required for inference using a batch size of 32 on the ModelNet40 dataset.

| Method | Inference Time |
|---|---|
| PointNet++ (Qi et al., 2017b) | 695.45 sec |
| Point Transformer (Zhao et al., 2021) | 701.88 sec |
| UWYN | 166.39 sec |

Table 9: **Batch inference:** This table shows the time required for inference using a batch size of 32 on the CIFAR 10 dataset.

| Method | Inference Time |
|---|---|
| LGViT (Xu et al., 2023) | 59.83 sec |
| AdaptFormer (Chen et al., 2022) | 62.18 sec |
| UWYN | 40.45 sec |

## A.6 MACS FOR OTHER NETWORKS

In this section we will compare the MACs of our networks with other popular transformer architectures. We are using the maximum MAC for our methods for comparison. From the table below, we demonstrate that our MACs are minimal with respect to other architectures as well.

Table 10: **MACs of other methods:** From the table, it is evident that our method is much more computationally efficient than the existing popular architecture choices. We use the thop (Developers, 2022) python library to calculate the MACs. Our MACs are very low compared to the other state of the art.

| Architecture | MACs |
|---|---|
| DeIT Small (Touvron et al., 2021b) | 4249 M |
| Swin V2 Tiny (Liu et al., 2021a) | 5760 M |
| Mobile ViT small (Mehta & Rastegari, 2022) | 347.5 M |
| EfficientNet-B0 (Tan & Le, 2019) | 380.55 M |
| ConvNeXt-T (Feng et al., 2022) | 4.5 G |
| MNv4-Hybrid-L (Qin et al., 2024) | 7.2 G |
| Our Complex Net (CIFAR 100) | 337 M |
| UWYN (CIFAR 100) | 1186 M |

## A.7 FEASIBILITY OF OUR EARLY EXIT

To explore if our concept of early exit is feasible or not, we have implemented the early exit from on a ViT (Kolesnikov et al., 2021). This indicates that after execution of each Transformer block, the output was sent to the classifier, and based on the classifier features and labels of the images, a cross-entropy loss was implemented. Table 11 demonstrates the results when we train these pipelines over a limited number of epochs, thereby testifying the feasibility of our work. We start the early exit after and confidence score calculation after at least 4 transformer blocks during inference.

Table 11: **Possibility of Early Exit:** Performance comparison of full capacity vs. early exit ViTs across various datasets, trained from scratch for 100 epochs. This table illustrates the generalizability of our method, showing that performance acceleration is consistent across different datasets and not solely reliant on the patch and attention head selector networks. From the full capacity accuracy (Acc.), there is a limited dip by 1%, while the other metrics, such as MACs, Parameters (Params), and Inference time (Time), have reduced significantly.

| Metric | CIFAR-10 (Full Capacity) | BloodMNIST (Full Capacity) | CIFAR-10 (Early Capacity) | BloodMNIST (Early Exit) |
|---|---|---|---|---|
| **Accuracy (%)** | 80.21 | 97.02 | 80.08 | 96.89 |
| **MACs (M)** | 1384 | 1401.8 | 836.6 | 596.4 |
| **Params (M)** | 21.31 | 21.31 | 12.34 | 12.34 |
| **Time (sec)** | 273.07 | 12.52 | 247.09 | 5.59 |

## A.8 OTHER MOTIVATIONAL EXAMPLES

As mentioned in the Introduction, Section 1, the redundancy in transformers is also apparent in Natural Language Processing tasks as well. We examine BART (Lewis et al., 2020), a widely used transformer model comprising 12 encoder and 12 decoder blocks, in the context of text classification task on the MNLI dataset (Williams et al., 2018). In this task, Lewis *et al*. (Lewis et al., 2020) categorizes if a pair of sentences as contradictory or not. We systematically drop later blocks in both the encoder and decoder to evaluate the impact on performance and computational efficiency. As Table 12 illustrates, reducing the number of blocks yields significant computational savings with only minor accuracy decreases. For instance, using 10 encoder blocks and 8 decoder blocks results in a mere 0.07% accuracy reduction while saving approximately 30 ms per CPU and GPU computation time, reducing FLOPs by almost 25%. These findings suggest a trade-off between accuracy and efficiency, which could be leveraged for applications where rapid inference is critical or resources are constrained.

Table 12: **Motivation from NLP:** Experimental results of BART (Lewis et al., 2020) on the MNLI dataset, demonstrating the impact of reducing model components (encoder/decoder blocks) on computational demand and accuracy.

| Encoders | Decoders | Params | CPU (ms) | GPU (ms) | KFLOPs | Accuracy (%) |
|----------|----------|--------|----------|----------|--------|--------------|
| 12 | 12 | $4.07 \times 10^8$ | 81.51 | 87.73 | $12.03 \times 10^3$ | 83 |
| 10 | 10 | $3.49 \times 10^8$ | 51.77 | 61.08 | $10.02 \times 10^3$ | 82.35 |
| 8 | 8 | $2.90 \times 10^8$ | 46.52 | 50.19 | $8.02 \times 10^3$ | 52.66 |
| 10 | 8 | $3.15 \times 10^8$ | 53.86 | 57.76 | $8.88 \times 10^3$ | 82.93 |
| 10 | 6 | $2.81 \times 10^8$ | 37.58 | 34.55 | $7.73 \times 10^3$ | 70.08 |

Table 12 highlights trade-offs between efficiency (CPU/GPU time for inference on the entire test set, parameters, KFLOPs) and classification performance, supporting the hypothesis that not all model components are essential for model efficiency. This indicates that intermediate features are also well learnt in most cases.

## B IMPLEMENTATION OF UWYN ON OBJECT DETECTION

We get the bounding boxes from SSD. From there, we resize the area inside the bounding boxes to meet the input requirement of UWYN (224,224). Then we train according to the image classification training procedure as described in Datasets and training under Section 4, Results of the main paper.

## C TRAINING THE HEAD AND PATCH SELECTORS

The model trains end-to-end using standard backpropagation despite performing seemingly discrete operations like head, token, and layer selection by leveraging differentiable approximations. Specifically, each selection module outputs logits that are passed through a Gumbel-Sigmoid (Binary Concrete) function, which samples continuous values in [0,1] that approximate Bernoulli decisions. During training, these soft masks modulate attention weights, feedforward layers, or token embeddings, allowing gradients to flow through the selection networks. In some cases, a straight-through estimator is applied, where the forward pass uses hard binary masks while the backward pass treats them as soft, maintaining differentiability. Temperature annealing gradually sharpens the mask distributions, encouraging the model to converge to near-binary selections, and optional sparsity regularization promotes pruning. At inference, the Gumbel noise is removed and deterministic thresholding is applied, enabling efficient hard pruning of heads, tokens, and layers. This approach allows the network to jointly learn which components are most informative for the task while optimizing the overall objective in a fully differentiable manner.