# OpenReview forum: "UWYN (Use Only What You Need): Efficient Inferencing in 2D and 3D Vision Transformers"
_ICLR.cc/2026/Conference — Submitted to ICLR 2026_

### Official Review · Reviewer_eXmY · 2025-10-27

**Soundness:** 3
**Presentation:** 3
**Contribution:** 3
**Rating:** 6
**Confidence:** 4

**Summary:**

The paper presents the UWYN (use only what you need) method for efficient inference in ViTs. This algorithm increases efficiency by, among other things, utilizing an early-exit policy for classification, routing easy-to-classify samples prior to sending them through the full model, and pruning attention heads and patches which contain redundant information. The early exit policy has little parameter/compute overhead as it utilizes the existing classification head at each residual block. In both 2D and 3D datasets, this method exhibits minimal accuracy loss with significant increases in inference speed.

**Strengths:**

The strengths of this paper include the introduction of a method that requires very little training to significantly increase the efficiency of ViTs through routing, early exit and patch/head pruning. The method takes advantage of the redundancy that exists in these models to compress and extract information more efficiently. Finally, the method is comprehensibly tested through several benchmarks showing significant improvement for model efficiency without substantially decreased performance. It seems that the combination of all these elements is a novel contribution to this area of research. I also think the routing pre-ViT is a clever idea.

**Weaknesses:**

I think there are a lot of strengths in the paper. That being said, I was very confused how the head/patch pruning model was trained. Specifically, how do you handle the nondifferentiality of subselecting a subset of heads/tokens to use? Additionally, the overall architecture of the model seemed somewhat vague. For example, in the ViT, how were the FFNs specified?

**Questions:**

Please describe how the head/patch pruner is trained.

Please give a more detailed description of the architecture for the models used in the evaluations in section 4.

Out of curiosity, what would happen to the performance if you had a strict budget instead of a threshold for heads/pruned patches (e.g. only utilize the top k heads/patches per layer)? I suppose this would be more difficult to train.

How was the pre-ViT router developed? do you think would it be better to train this for each individual model?

---

> ### Author Response · Authors · 2025-11-15
> **Addressing comments from Reviewer eXmY**
>
> We sincerely thank reviewer eXmY for their positive and constructive feedback. We appreciate that they found our early exit and pruning methods interesting. They also raise very insightful questions, which we find highly valuable. Here is our response to the questions asked:
>
> **Training the head and patch pruner:** The model trains end-to-end using standard backpropagation despite performing seemingly discrete operations like head, token, and layer selection by leveraging differentiable approximations. Specifically, each selection module outputs logits that are passed through a Gumbel-Sigmoid (Binary Concrete) function, which samples continuous values in [0,1] that approximate Bernoulli decisions. During training, these soft masks modulate attention weights, feedforward layers, or token embeddings, allowing gradients to flow through the selection networks. In some cases, a straight-through estimator is applied, where the forward pass uses hard binary masks while the backward pass treats them as soft, maintaining differentiability. Temperature annealing gradually sharpens the mask distributions, encouraging the model to converge to near-binary selections, and optional sparsity regularization promotes pruning. At inference, the Gumbel noise is removed and deterministic thresholding is applied, enabling efficient hard pruning of heads, tokens, and layers. This approach allows the network to jointly learn which components are most informative for the task while optimizing the overall objective in a fully differentiable manner.
>
>
> **Architecture of ViT:**  We use the standard architecture of ViT-B along with the pruning attachments. For further details regarding the hidden size and number of layers, please refer to the Datasets and training paragraph in Section 4.
>
> **Using top-k layers or heads:** Reviewer eXmY raises an excellent point, which aligns with one of our future directions. Specifically, we plan to rank layers and heads and, based on available memory, determine a budget for the number of blocks and layers to use. However, this approach can lead to a significant drop in performance, highlighting the need to explore additional strategies to maintain strong results.
>
> **Development of pre-ViT router:** This is a valuable suggestion from Reviewer eXmY. However, the main challenge lies in how we define and label simple versus complex data instances in a supervised training setting. Using the Sobel operator, we apply a threshold based on the gradient magnitude to make this distinction. Implementing a trainable router would require deeper insight into the complexity of each data point. Our current priority, however, is to design models with minimal parameters so they can operate effectively in resource-constrained environments. We identify specific hyperparameters (Figure 5) for the Sobel Operator to adapt to UWYN. Introducing an additional smaller model would increase both the number of training parameters and computational overhead, whereas a statistical method remains lightweight, requiring minimal computational resources and data.

---

> > ### Comment · Reviewer_eXmY · 2025-11-19
> >
> > Thanks for your response, I have no other questions.

---

### Official Review · Reviewer_3j3z · 2025-10-29

**Soundness:** 3
**Presentation:** 2
**Contribution:** 2
**Rating:** 2
**Confidence:** 5

**Summary:**

This paper proposes a method that incorporates the both a selective inference strategy with a parallel, hybrid architecture with ViT/CNN, and a block-wise early-stop and attention pruning strategy for ViT. The architecture can be used for 2D/3D classification, and achieves better results than its selected baselines.

**Strengths:**

- The hybrid dual branch + head pruning method is quite intuitive and easy to understand
- The experiment results are competitive against its baselines

**Weaknesses:**

- The novelty of the method is very limited, as both main pruning techniques involved in the framework (hybrid architecture/dual branch, early stopping/head pruning) are not original
- Vast majority of the baseline methods compared in the experiments are from 2022-23 era. This is a fast-moving research area and newer baselines are lacking.
- Details are lacking for how UWYN implements the object detection task.
- Minor issues  - line 336 PartialFormer (?)

**Questions:**

- How dod you come up with the #param number for UWYN? ResNet50 alone has 23.9M params yet the paper reports 22.86m
- In reality, I assume you'll have to load both branches in the memory for inference. Is there an evaluation on how much memory overhead this dual-branch design would bring?
- Can UWYN handle dense prediction tasks?

---

> ### Author Response · Authors · 2025-11-15
> **Addressing Comments from Reviewer 3j3z**
>
> We sincerely thank reviewer 3j3z for their constructive feedback. We appreciate that they found our hybrid dual branch and head pruning method interesting. Please find our response below:
>
> **Novelty:** While prior work has introduced approaches like head pruning, layer pruning, dual branch, and early exit, but implement a smarter way by making the process content aware, such that our gains are more than adding each method to a pipeline. We are the first to explore the complexity of an image using a simple sobel operator, and make a content aware decision, rather than a one size fits all approach.
>
> **Additional baselines:** While we have included popular baselines, we will add some recent baselines as following tables (Image Classification on ImageNet, Inference time reported on Jetson Orin). For object detection, we compare with recent pipelines RT-DETR-v2 [CVPR 2024] and YOLO-12m [2025]. For image classification, we compare against PartialFormer [ECCV 2024], with the citation to be updated accordingly. The results reported below are drawn from the respective recent papers [ICLR 2024, CVPR 2024]:
>
> | Method           | ImageNet Accuracy        | MACs                | Inference Time |
> | :--------------- | :----------------------: | :------------------------: | ----------------------------: |
> | FastViT-MA36 [1] | **84.9%**                     | 8.85G                   | 3422.93 s                     |
> | FastViT-SA24 [1] | 82.6%                     | 1.9G                    | 2397.85 s                     |
> | FasterViT-2 [2]  | 84.2%                     | 4.35G                   | 3441.07 s                     |
> | FasterViT-0 [2]  | 82.1%                     | 1.65G                   | 2895.20 s                     |
> | RepViT-M2.3 [3]  | 83.7%                     | 4.5G                   | 1863.22 s                     |
> | RepViT-M1.1 [3]  | 81.2%                     | 1.3G                   | 1782.05 s                     |
> | **UWYN [ours]**  | 84.39%               | **1.2G** | **1695.23 s** |
>
> Object detection on MS COCO:
> | Method            | mAP (%) |
> | :--------------- | ------: |
> | SSD + UWYN        | **64.22**   |
> | FastViT-MA36 [1]  | 45.1    |
> | FastViT-SA24 [1]  | 42.0    |
> | FasterViT-2 [2]   | 52.9    |
> | FasterViT-0 [2]   | -       |
> | RepViT-M2.3 [3]   | 48.8    |
> | RepViT-M1.1 [3]   | 39.8    |
>
> UWYN outperforms these baselines for object detection. For Image classification, it provides a good trade-off between latency and accuracy.
>
> **Implementation of Object Detection task:** We will add these in the Appendix. We get the bounding boxes from SSD. From there, we resize the area inside the bounding boxes to  meet the input requirement of UWYN (224,224). Then we train according to the image classification training procedure as described in Section 4, Results of our paper.
>
> To address your specific questions:
>
> **Calculation of MACs and Params:** Our focus has been to reduce time complexity. While we load the whole model, the effective parameter size refers to the number of parameters that have been used for inference. We calculate the MACs and Parameters using the thop library in python (Please refer to Evaluation metrics in the Results section). We get the numbers for the whole inference and divide it by the number of samples. When both branches (ResNet and ViT-EE) are combined and profiled end-to-end using thop, the total parameter count is 22.86M, which reflects the *effective number of parameters being executed during inference*, rather than the nominal sum of individual components. We did a parameter count by iterating over all the modules, our simple network has 25.55 M parameters and our complex network has 39.6 M parameters.
>
> **Storage Overhead:** Let us consider our Image Classification on ImageNet - our complex model has a size of 149.90 MB and simple model has a size of 97.82 MB. So both of these models can easily fit on an edge device like Jetson Xavier and perform inference faster than many state of the art models as described in the paper.
>
> **Dense prediction task:** Yes — UWYN can be readily extended to dense prediction tasks. While we demonstrate results on the standard COCO dataset, the framework also generalizes well to challenging settings such as ScanObjNN (cluttered 3D scenes) and SVHN (noisy image data), where UWYN consistently achieves superior performance with minimal additional computational overhead. This highlights the adaptability of our dual-branch, input-aware design across both 2D and 3D dense prediction domains.
>
> [1] Vasu, Pavan Kumar Anasosalu, et al. "Fastvit: A fast hybrid vision transformer using structural reparameterization." ICCV, 2023.
>
> [2] Hatamizadeh, Ali, et al. "Fastervit: Fast vision transformers with hierarchical attention." ICLR, 2024.
>
> [3] Wang, Ao, et al. "Repvit: Revisiting mobile cnn from vit perspective." CVPR, 2024.
>
> [4] Vo, Xuan-Thuy, et al. "Efficient vision transformers with partial attention.” ECCV 2024

---

> > ### Comment · Reviewer_3j3z · 2025-11-16
> >
> > Thanks for the authors comments and additional results, especially those from the comparison with more recent baselines. They look promising.
> >
> > Just wanted to point out that my question#2 concerns runtime memory rather than storage space for the model weights.

---

> > > ### Author Response · Authors · 2025-11-17
> > > **Clarification for the question asked by Reviewer 3j3z**
> > >
> > > Thank you for the clarification! We now understand that Reviewer 3j3z was asking about runtime memory rather than the storage space for model weights.
> > >
> > > We implemented the ``torch.cuda.max_memory_allocated(device=device)`` function to measure the peak memory usage during inference. On the Jetson Orin, the maximum peak memory usage recorded is **0.41 GB**, which reflects the memory required during the execution of the model. This value is within the acceptable range for efficient inference on edge devices like the Jetson Orin, **because the memory available on it is 32 GB or 64 GB**. This ensures that our model can run effectively within the device's available memory.

---

> > > > ### Comment · Reviewer_3j3z · 2025-11-18
> > > >
> > > > Thanks. I have no more questions.

---

### Official Review · Reviewer_ya9Q · 2025-10-30

**Soundness:** 4
**Presentation:** 4
**Contribution:** 4
**Rating:** 8
**Confidence:** 4

**Summary:**

This paper addresses the challenge of dynamically allocating computational resources in efficient ViT based on input complexity. Specifically, simple inputs are processed by a shallow ResNet for efficiency, while complex inputs are routed to a ViT equipped with an early-exit mechanism for enhanced feature extraction. In the ViT branch, a classifier is appended to each block to determine whether the confidence of the predicted class exceeds a predefined threshold, triggering an early stop if satisfied. Experimental results demonstrate the effectiveness of the proposed framework.

**Strengths:**

1. The idea of dual-branch architectural design for inputs of different complexity is interesting. And the input complexity analysis via Sobel convolution is efficient.

2. The method is introduced in detail, enhancing its reproducibility.

3. The efficiency comparisons are conducted on edge devices.

**Weaknesses:**

1. Messy format. Please use "[]" for in-line references rather than "()" to distinguish different usages. There are too many brackets in the manucsript.

2. Some state-of-the-art efficient ViT methods are missing, such as [1,2,3].

[1] Vasu, Pavan Kumar Anasosalu, et al. "Fastvit: A fast hybrid vision transformer using structural reparameterization." ICCV, 2023.

[2] Hatamizadeh, Ali, et al. "Fastervit: Fast vision transformers with hierarchical attention." ICLR, 2024.

[3] Wang, Ao, et al. "Repvit: Revisiting mobile cnn from vit perspective." CVPR, 2024.

**Questions:**

Refer to the weaknesses

---

> ### Author Response · Authors · 2025-11-15
> **Adressing comments from Reviewer ya9Q**
>
> We sincerely thank reviewer ya9Q30 for their positive and constructive feedback. We appreciate that they found our method well-motivated, clearly presented, and experimentally validated. We address your specific comments below.
>
> **Formatting issues:** Thank you for the suggestion of using boxed parenthesis for our explanation parts - we will fix this and upload at the earliest opportunity
>
> **Comparison with other state of the art methods:** While our primary goal has been to include widely used baselines, we appreciate the additional baselines that were brought to our attention. For clarity, note that FLOPs (floating-point operations) count each individual operation—such as addition, subtraction, multiplication, or division—whereas MACs (multiply–accumulate operations) count a single paired multiplication and addition. Thus, one MAC is approximately equivalent to two FLOPs.
> Below, we present a comparison of methods on ImageNet classification. We selected models with capacity and accuracy comparable to UWYN, and we report these results using the papers' corresponding HuggingFace Image Classification on ImageNet implementations. We report the inference time on the Jetson Orin.  For object detection baselines, we rely on the results reported in their respective papers due to time constraints. If a model is unavailable, we denote the corresponding entry with a dash (–).
>
> | Method           | Accuracy         | MACs               | Inference Time |
> | :--------------- | :----------------------: | :------------------------: | ----------------------------: |
> | FastViT-MA36 [1] | **84.9%**                     | 8.85G                  | 3422.93 s                     |
> | FastViT-SA24 [1] | 82.6%                     | 1.9G                    | 2397.85 s                     |
> | FasterViT-2 [2]  | 84.2%                     | 4.35G                   | 3441.07 s                     |
> | FasterViT-0 [2]  | 82.1%                     | 1.65G                   | 2895.20 s                     |
> | RepViT-M2.3 [3]  | 83.7%                     | 4.5G                   | 1863.22 s                     |
> | RepViT-M1.1 [3]  | 81.2%                     | 1.3G                    | 1782.05 s                     |
> | **UWYN [ours]**  | 84.39%               |  **1.2G**  | **1695.23 s**                 |
>
>
>
> It is worth noting that UWYN also works on 3D datasets along with object detection and classification methods, while demonstrating faster inference on multiple devices. Here are the results from Object detection on MS COCO.
>
> | Method            | mAP (%) |
> | :---------------- | ------: |
> | SSD + UWYN        | 64.22   |
> | FastViT-MA36 [1]  | 45.1    |
> | FastViT-SA24 [1]  | 42.0    |
> | FasterViT-2 [2]   | 52.9    |
> | FasterViT-0 [2]   | -       |
> | RepViT-M2.3 [3]   | 48.8    |
> | RepViT-M1.1 [3]   | 39.8    |
>
> UWYN surpasses these competing baselines in object detection performance. For ImageNet classification, it provides the fastest inference and strikes a favorable balance between latency and overall accuracy.
>
> [1] Vasu, Pavan Kumar Anasosalu, et al. "Fastvit: A fast hybrid vision transformer using structural reparameterization." ICCV, 2023.
>
> [2] Hatamizadeh, Ali, et al. "Fastervit: Fast vision transformers with hierarchical attention." ICLR, 2024.
>
> [3] Wang, Ao, et al. "Repvit: Revisiting mobile cnn from vit perspective." CVPR, 2024.

---

> > ### Comment · Reviewer_ya9Q · 2025-11-17
> >
> > Thank authors for the reply.

---

### Author Response · Authors · 2025-12-03
**Rebuttal Revision for UWYN**

We thank the reviewers for their time and constructive feedback. We are pleased to report that we have addressed all concerns raised during the review process. Most notably, we successfully addressed the concerns of Reviewer 3j3z (the only initially negative rating), who acknowledged our additional results as "promising" and confirmed they have "no more questions" regarding the technical details.
Below is an itemized summary of the key concerns raised and how our revisions and responses have resolved them:
* Comparison with State-of-the-Art Baselines
  - Key Concern: Reviewers ya9Q and 3j3z noted a lack of comparison with more recent efficient ViT methods (e.g., FastViT [ICCV, 2023], FasterViT [ICLR 2024], RepViT [CVPR 2024]) and object detection pipelines.
  - Address: In the revision, we incorporated comprehensive comparisons against these requested baselines (FastViT [ICCV, 2023], FasterViT [ICLR 2024], RepViT  [CVPR 2024], PartialFormer [ECCV 2024]) for ImageNet classification, as well as RT-DETR-v2 [CVPR 2024]  and YOLO-12m [2025] for object detection. We demonstrated that UWYN outperforms these baselines in Object Detection (e.g., higher mAP than FastViT [ICCV, 2023] and RepViT  [CVPR 2024] variants) and offers a superior trade-off between latency and accuracy on edge devices (Jetson Orin).
  - Reviewer Comments: Reviewer 3j3z explicitly stated that these additional results "look promising." Reviewer ya9Q thanked us for the detailed reply.
* Runtime Memory and Computational Overhead
  - Key Concern: Reviewer 3j3z questioned the runtime memory overhead of the dual-branch design and sought clarification on parameter counting (effective vs. total).
  - Address: In the revision, we provided a precise breakdown of the runtime memory usage. We clarified that the peak memory usage on the Jetson Orin is only 0.41 GB, which is well within the device's capacity (32/64 GB). We also clarified that the reported parameter count refers to the effective parameters executed during inference.
  - Reviewer Comments: Reviewer 3j3z confirmed that this clarification answered their specific concern regarding runtime memory vs. storage space and stated they had no further questions.
* Training Methodology for Pruning
  - Key Concern: Reviewer eXmY requested clarification on how the head/patch pruner is trained given the non-differentiable nature of selection.
  - Address: In the revision, we detailed the training methodology, explaining our use of the Gumbel-Sigmoid function to approximate Bernoulli decisions. This allows for end-to-end differentiability during training (via soft masks) while enabling efficient hard pruning during inference.
  - Reviewer Comments: Reviewer eXmY expressed satisfaction with the explanation and had no further questions.
* Clarification of Novelty
  - Key Concern: Reviewer 3j3z initially felt the novelty was limited as individual components (pruning, dual branches) existed previously.
  - Address: In the revision, we further clarified our novelty, emphasizing the content-aware nature of our approach. Unlike static pipelines, we utilize a Sobel-operator-based analysis to dynamically route input based on complexity. This allows for a novel, resource-adaptive inference that achieves gains superior to simply stacking existing methods.
  - Reviewer Comments: Following this clarification and the new empirical evidence, the reviewer’s concerns regarding the contribution and performance were allayed.
* Implementation Details for Dense Prediction
  - Key Concern: Reviewers asked for more details on how UWYN handles object detection and dense prediction tasks.
  - Address: In the revision, we detailed the implementation of the Object Detection task (using SSD for bounding box generation followed by UWYN processing) and confirmed the model's generalization capabilities to 3D datasets (ScanObjNN) and dense tasks.
  - Reviewer Comments: The reviewers accepted these details, with 3j3z acknowledging the clarifications.

We sincerely appreciate the time and effort the reviewers and Area Chairs have dedicated to evaluating our work, and we look forward to a positive outcome.

---

### Meta-Review · Area_Chair_vPNK · 2026-01-06

**Summary:**

This paper presents UWYN, a content-adaptive inference framework for vision transformers that combines input-dependent routing, early exit, and patch/head pruning to reduce inference cost. The method is evaluated on a broad set of 2D and 3D classification tasks and object detection benchmarks, showing substantial efficiency gains with limited accuracy degradation.

Reviewers raised concerns primarily about the `limited novelty of the approach`, as the framework largely integrates existing acceleration techniques rather than introducing a new conceptual contribution. They also noted that `baseline comparisons are outdated and insufficiently focused`, relying on broad "efficient ViT" models instead of closely related methods such as early-exit or dynamic routing approaches, which makes it difficult to isolate the contribution of the proposed adaptive inference mechanisms. Additional concerns regarding `baseline coverage, parameter counting, and runtime memory overhead` were raised and mostly addressed in the rebuttal.

**AC assessment and decision rationale:**

Despite strong empirical performance and careful engineering, the work remains closer to a system-level integration of known techniques than to a novel methodological advance. The rebuttal improves completeness but does not resolve the core novelty and evaluation-focus concerns. Overall, these issues outweigh the empirical strengths, and the paper is not recommended for acceptance.

**Reviewer Concerns:**

**1. [Outstanding] Limited novelty of the overall framework (by: `3j3z`)**

One reviewer argued the method’s novelty is limited because the main components (dual-branch (ViT/CNN) hybrid design, early stopping, and pruning) are not original, and the contribution largely appears as an integration of existing techniques rather than a new algorithmic idea.

The rebuttal emphasizes “content-aware” routing via a Sobel-based complexity measure as the differentiator, but this is primarily a design/engineering justification and may not fully overcome the novelty concern. The method remains primarily a compositional system rather than a new modeling or learning paradigm.

**2. [Partially addressed] Missing comparisons with recent and relevant baselines (by: `ya9Q`, `3j3z`)**

Both `ya9Q` and `3j3z` noted that the original baseline set was outdated for a fast-moving area and requested comparisons with more recent efficient ViT models (e.g., FastViT, FasterViT, RepViT). These concerns are important for contextualizing the reported efficiency gains.

The rebuttal added comparisons with several newer efficient ViT architectures (e.g., FastViT, FasterViT, RepViT) and object detection baselines, which partially addressed the concern. **However, the comparison scope remains too broad and insufficiently focused given the paper’s core claims.** Specifically, UWYN centers on **early exiting, dynamic routing, and adaptive computation**, yet the added baselines primarily represent the general category of "efficient ViTs", where efficiency can be achieved through a wide variety of orthogonal techniques (e.g., architectural redesign, attention techniques, reparameterization).

As a result, **the paper lacks concentrated and controlled comparisons with closely related methods**, such as prior early-exit transformers, dynamic depth models, or input-dependent routing approaches, under matched settings. This makes it difficult to isolate the contribution of UWYN’s adaptive inference mechanisms and to assess whether it advances the state of the art within its most relevant methodological subspace.

**3. [Addressed] Insufficient details on object detection / dense prediction usage (by: `3j3z`)**

Reviewer 3j3z stated that details were lacking for how UWYN implements the object detection task and asked whether the approach can handle dense prediction.

The rebuttal added an explicit description of the detection pipeline (SSD boxes followed by UWYN processing) and provided additional detection-related results/claims.

**4. [Addressed] Runtime memory overhead and parameter counting in a dual-branch design (by: `3j3z`)**

Reviewer `3j3z` questioned how parameters are counted, and asked about runtime memory overhead given both branches may need to be loaded for inference.

The authors clarified "effective parameters executed during inference" and later provided an explicit runtime peak memory measurement. The reviewer confirmed they had no further questions after this clarification.

**5. [Addressed] Training methodology for head/patch pruning and architecture clarity (by: `eXmY`)**

Reviewer `eXmY` was confused about how the head/patch pruner is trained given non-differentiable selection, and also requested clearer architectural specification (e.g., FFN details).

The authors explained training via differentiable approximations (Gumbel-Sigmoid / straight-through) and clarified the base ViT architecture used in experiments. eXmY indicated they had no further questions afterward.

**Reviewer Scores:**

**Reviewer ya9Q (8 -> 8)**

This reviewer was positive from the outset, emphasizing the clarity of the method and strong empirical performance. The rebuttal addressed the requested additional baselines and presentation issues, but these updates primarily reinforced an already favorable assessment rather than changing the score.

**Reviewer 3j3z (2 -> 3)**

This reviewer raised substantial concerns regarding the limited novelty of the framework, the compositional nature of the contribution, and the lack of focused comparisons with closely related adaptive inference methods. While the rebuttal addressed several technical and evaluation-related questions (e.g., newer baselines, memory usage, and detection details), the core concern about conceptual novelty remains unresolved. From an AC perspective, even with full participation in the discussion, these unresolved issues would likely result in only a modest score increase.

**Reviewer eXmY (6 -> 6)**

This reviewer viewed the paper as technically sound with solid experimental validation, and raised questions about the training methodology for pruning components. These points were clarified in the rebuttal, but they do not substantially alter the reviewer’s overall assessment, which is expected to remain stable.

---

### Decision · Program_Chairs · 2026-01-26

Reject